# Sensorless Control Method of High-Speed Permanent Magnet Synchronous Motor Based on Discrete Current Error

Zhiqiang Wang [1], Dezheng Du [1], Qi Yu [2], Haifeng Zhang [2], Chen Li [3], Liyan Guo [3], Xin Gu [1] and Xinmin Li [1,*]

1 School of Electrical Engineering, Tiangong University, Tianjin 300387, China
2 Baodi Power Supply Branch of State Grid Tianjin Electric Power Company, Tianjin 301800, China
3 Advanced Electrical Equipment Innovation Center, Zhejiang University, Hangzhou 311107, China
* Correspondence: lixinmin@tiangong.edu.cn

**Abstract:** When the Surface-Mounted Permanent Magnet Synchronous Motor (SPMSM) is in the condition of high-speed and low carrier-wave ratio, the performance of sensorless control is more affected by digital control delay, parameter inaccuracy, and other factors. This paper presents a sensorless control method based on the static error between the discrete *d*-axis current and the corresponding reference value. If there is no error in position angle, the discrete *d*-axis current should have no static error approximately. In addition, the static error of the *d*-axis current is related to the speed, so the PI controller with a variable proportional integral coefficient is used to ensure a stable error compensation performance in a wide speed range. The proposed method can accurately compensate the estimated rotor position of the motor under high-speed and low carrier ratio conditions and improve the accuracy of sensorless control. It provides an effective measure for the stable and reliable acceleration of electric vehicles and has specific practical significance for the development of electric vehicle control.

**Keywords:** high-speed permanent magnet synchronous motor; sensorless control; angle compensation; low carrier-wave ratio; discrete current error





## 1. Introduction

Surface-mounted permanent magnet synchronous motor has the advantages of simple structure, high power density, and strong reliability. It is widely used in industrial robots, aerospace, and especially electric vehicles. For permanent magnet synchronous motor control, the rotor position information has an important influence on the control performance of the system [1–5]. However, on some special occasions, it is difficult to install a position sensor in the motor system due to factors such as installation volume and speed. As a result, the permanent magnet motor needs to adopt a sensorless control method [6]. When the motor is running in the high-speed area, the back electromotive force (EMF) is relatively large. At this time, the back EMF can be observed to obtain more accurate rotor position information. Commonly used observation methods include sliding mode observer [7], model reference adaptation [8], Lomberg Observer [9,10], etc. Among them, sliding mode observers are widely used due to their robustness and simple algorithm [11,12].

In the actual experimental system, the switching frequency of the power device is determined. Under this constraint, when the motor enters the high-speed region, inevitably, the ratio of the switching frequency to the motor rotor frequency is low; that is, the carrier ratio is low. Under this condition, the inherent delay, dead zone effect, and current harmonics of digital control will increase [13,14]. When the traditional back-EMF observation method is used to control the motor at high-speed without position control, the error between the position angle estimated by the observer and the actual angle will be further increased, resulting in a rapid decrease in current quality, which in turn will generate greater torque fluctuations, forming a vicious circle for motor control [15,16].

In [17], the three-phase current is sampled multiple times, and the calculation delay is estimated, and then the three-phase current is compensated to eliminate the calculation delay. This method mainly compensates for the digital control delay, The disadvantage is that the dynamic performance is poor. In the literature [18], the magnitude of the three-phase current amplitude in the steady state is used as the basis for the existence of the angle error, and the rotor position angle estimated when the motor is running stably is compensated. When the three-phase current amplitude is the smallest, the compensation is considered. The angle is the most accurate. This method requires that the motor operating conditions do not change before and after the angle compensation, the compensation speed of this method is slow, and the dynamic performance is insufficient. In [19], currents with different amplitudes are injected into the *d*-axis four times. The motor parameter identification method is used, and the negative impact caused by the inverter nonlinearity is analyzed. However, the identification time is too long, and additional signal injection will also cause interference.

This article first introduces the principle of using a sliding mode observer to estimate the back EMF to obtain the position angle and speed of the motor rotor; then, based on the algorithm, the main factors leading to the error in the position angle are analyzed, these factors mainly include the inherent phase delay caused by the sensorless control method itself due to the filter, the error caused by the change of the motor parameters, and the error caused by the dead zone effect. Aiming at the above-mentioned angle error, this paper proposes a sensorless control method based on the discrete *d*-axis current static difference, which can effectively eliminate the above-mentioned angle error. The principle of this method is to use the static difference between the discrete *d*-axis current and the given value of the *d*-axis current as the input of the proportional-integral (PI) control algorithm, from which the static difference is calculated to obtain the angle error value; then the value is compensated to the estimated rotor position. When the angle error compensation is completed, theoretically the static difference of the *d*-axis current should be approximately zero. This method accurately compensates the estimated rotor position angle, makes the output value of coordinate transformation more accurate, and improves the accuracy and performance of sensorless control, avoiding the motor being out of control due to large position errors during the acceleration process. Finally, this paper verifies the feasibility and effectiveness of the proposed method through simulation and experiment.

## 2. Rotor Position Estimation Method and Angle Error Analysis Based on Sliding Mode Observer

### 2.1. Principle of Estimating Rotor Position Based on Sliding Mode Algorithm

In this paper, a high-speed surface-mount permanent magnet synchronous motor is used as the research object. The voltage equation of the surface-mount permanent magnet synchronous motor in the static coordinate system is shown in the following Formula [20]:

$$\begin{cases} \frac{\mathrm{d}i_\alpha}{\mathrm{d}t} = -\frac{R}{L}i_\alpha + \frac{1}{L}u_\alpha - \frac{1}{L}e_\alpha \\ \frac{\mathrm{d}i_\beta}{\mathrm{d}t} = -\frac{R}{L}i_\beta + \frac{1}{L}u_\beta - \frac{1}{L}e_\beta \end{cases} \tag{1}$$

where $u_\alpha$ and $u_\beta$ are the motor stator terminal voltage and shaft component, respectively; $i_\alpha$ and $i_\beta$ are the $\alpha$ and $\beta$ axis components of the motor current, respectively; $e_\alpha$ and $e_\beta$ are $\alpha$ and $\beta$ axis components of the motor back electromotive force; $R$ is the stator resistance, and $L$ is the motor inductance.

The back electromotive force components $e_a$ and $e_b$ of the motor can be expressed as:

$$\begin{cases} e_\alpha = -\omega_e\psi_f\sin\theta \\ e_\beta = \omega_e\psi_f\cos\theta \end{cases} \tag{2}$$

where $\omega_e$ is the electromechanical angular velocity, $\psi_f$ is the motor rotor flux linkage, $\theta$ is the motor rotor position angle.

The following Formula (3) is the sliding mode observer formula based on Formula (1):

$$
\begin{cases}
\dfrac{\mathrm{d}\hat{i}_\alpha}{\mathrm{dt}} = -\dfrac{R}{L}\hat{i}_\alpha + \dfrac{1}{L}(u_\alpha - z_\alpha) \\[2mm]
\dfrac{\mathrm{d}\hat{i}_\beta}{\mathrm{dt}} = -\dfrac{R}{L}\hat{i}_\beta + \dfrac{1}{L}(u_\beta - z_\beta)
\end{cases}
\tag{3}
$$

where $\hat{i}_\alpha$ and $\hat{i}_\beta$ are the estimated values of $i_\alpha$ and $i_\beta$ respectively, $z_\alpha$ and $z_\beta$ and are respectively the $\alpha$ and $\beta$ axis current error switch signal, and the expression is:

$$
\begin{cases}
z_\alpha = K_s \mathrm{sgn}(\hat{i}_\alpha - i_\alpha) \\[2mm]
z_\beta = K_s \mathrm{sgn}(\hat{i}_\beta - i_\beta)
\end{cases}
\tag{4}
$$

where $K_s$ is the sliding mode gain, and sgn is the sign function. Since $z_\alpha$ and $z_\beta$ are high-frequency switching signals containing back-EMF information, they need to be passed through a low-pass filter to get the back-EMF estimates $\hat{e}_\alpha$ and $\hat{e}_\beta$; coupled with the quadrature phase-locked loop, the rotor position $\hat{\theta}$ and speed $\hat{\omega}_e$ can be calculated. The principle block diagram of the rotor position estimation method based on the sliding mode observer is shown in Figure 1.

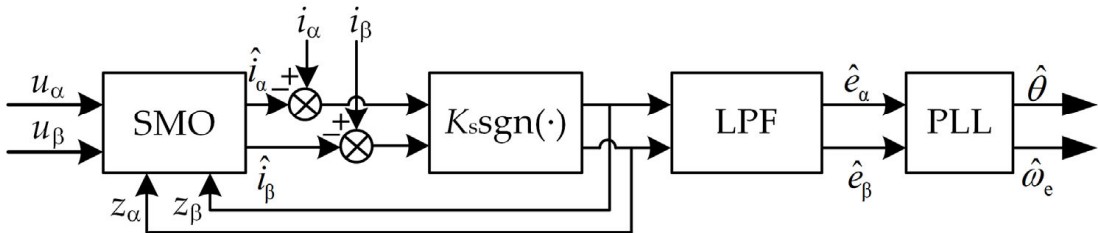

**Figure 1.** Rotor position estimation method based on sliding mode observer.

### 2.2. Analysis of Estimation Angle Error in the State of High-Speed and Low Carrier Wave Ratio

When the permanent magnet synchronous motor system is in a low carrier wave ratio operating condition, various factors include the delay of the sensorless control algorithm itself, the inherent delay of digital control, the dead zone effect, and the phase delay caused by the nonlinearity of the inverter, etc. The error between the estimated angle and the actual angle of the position control algorithm increases, which seriously affects the motor control performance.

#### 2.2.1. The Delay of the Sensorless Algorithm

It can be seen from Figure 1 that in order to filter out the high-frequency signals in $z_\alpha$ and $z_b$, the low-pass filter is an important part of the sensorless algorithm, but the signal will have a certain phase delay after passing through the filter. Therefore, it is necessary to add angle compensation to the traditional sensorless algorithm to compensate for the angle delay caused by the low-pass filter. The specific compensation Formula is:

$$
\theta_{\mathrm{LPF}} = \arctan(\hat{\omega}_e / \omega_c)
\tag{5}
$$

where $\omega_e$ is the cut-off frequency of the low-pass filter.

In addition, the sliding mode observer method is a position estimation method based on the fundamental wave model of the motor, which will be affected by changes in motor parameters to a certain extent. Under high-speed operating conditions, the motor inductance parameters change with the saturation of the magnetic field, resulting in a difference between the nominal motor parameters used in the position estimation process and the actual parameter values. At this time, the calculated position angle will also have a specific error [21,22].

### 2.2.2. Digital Control Delay

In the experiment, a switching frequency of 4 kHz is used, the number of pole pairs of the motor is 2, the operating speed is 9000 r/min, and the carrier ratio is 13. At this point, the motor is in a low carrier ratio, and the impact of digital control delay is more obvious. The traditional sampling mode is shown in Figure 2.

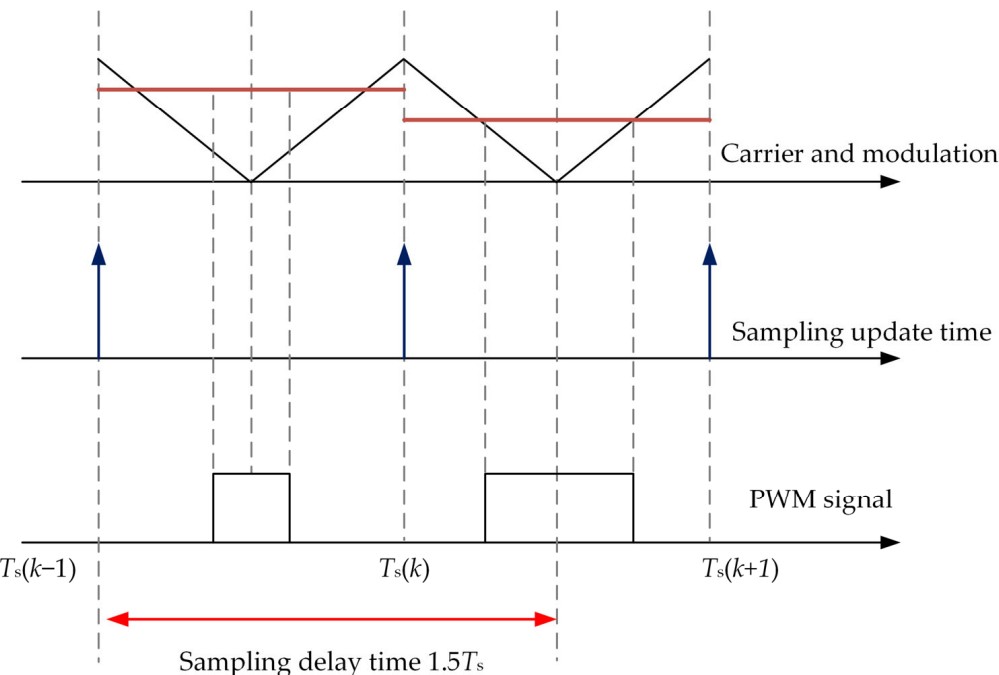

$T_s(k−1)$        $T_s(k)$        $T_s(k+1)$

Sampling delay time $1.5T_s$

**Figure 2.** Traditional sampling and PWM update method.

In the figure, $T_s$ is the control period, which is equal to the sampling period and the switching period at the same time. The three-phase current is sampled at $T_s(k − 1)$, and then the duty cycle signal is calculated for a certain period of time. The update can only be performed at $T_s$. Therefore, from the start of the control algorithm to the beginning of the controller to update the pulse width modulation (PWM) output, there is a Control cycle delay.

In addition, it takes half a switching cycle to apply the calculated PWM signal to the switching tube before it can be considered that the average value of the inverter output voltage is equal to the reference voltage. Therefore, from the start of the algorithm to the application of PWM to the inverter, the total delay of the digital control system is:

$$T_{\text{delay}} = 1.5T_{\text{PWM}} \tag{6}$$

Under high-speed and low-carrier ratio conditions, the impact of the above-mentioned digital delay is more serious, so angle compensation is necessary [23].

### 2.2.3. Other Delay Issues

In order to prevent the inverter from passing through the upper and lower bridge arms in the same phase, a dead zone needs to be added to the PWM signal, and the addition of the dead zone increases the deviation between the actual output voltage and the reference voltage, which is also an important cause of current distortion [24].

In the sensorless algorithm, the distortion of the voltage and current in the stationary coordinate system will also cause the calculated position angle error to increase, especially in the high-speed and low-carrier ratio conditions, the above-mentioned error is more prominent.

In addition to the above delays, there are some phase delays that are difficult to quantize, including sampling delay, analog-to-digital (AD) conversion delay, etc., [25,26]. Under high-speed and low carrier wave ratio conditions, the effects of these delays will also be amplified, and the estimated rotor position angle will also have a non-negligible delay angle, thereby reducing the quality of current loop control.

### 3. Angle Compensation Method Based on Discrete Current Error

Aiming at the unique problems of motor position control under high-speed and low carrier ratio conditions, this article first uses double sampling and double update mode to increase the current control frequency in one switching cycle and then control the algorithm output reference voltage and space vector pulse width modulation (SVPWM) duty cycle in one switching cycle Two updates are made to improve the reference voltage tracking accuracy and current loop control performance in the high-speed and low carrier wave ratio state without increasing the switching frequency; secondly, in the above control mode, a high-speed and low-speed based on discrete current error is proposed. There is a sensorless control algorithm under carrier ratio to realize the estimation of rotor position angle compensation.

#### 3.1. Analysis of Angle Error Judgment Basis

In this paper, the speed and current double closed-loop control architecture are adopted. The *q*-axis current reference value is obtained through the speed loop PI controller. The *d*-axis is controlled by $i_d = 0$ at the base speed, and the voltage feedback method is used for the field weakening control above the base speed. It can be seen that, in theory, if the model in the control system is accurate and the rotor position angle has no error, there should be no static difference between the *d*-axis current and the *d*-axis reference value. Therefore, in this paper, the presence or absence of the static error of the *d*-axis current discretized by the Euler approximation method is used as the basis for judging whether there is an error in the estimated angle in the position control. The control method proposed in this paper is shown in Figure 3. The following is a theoretical analysis.

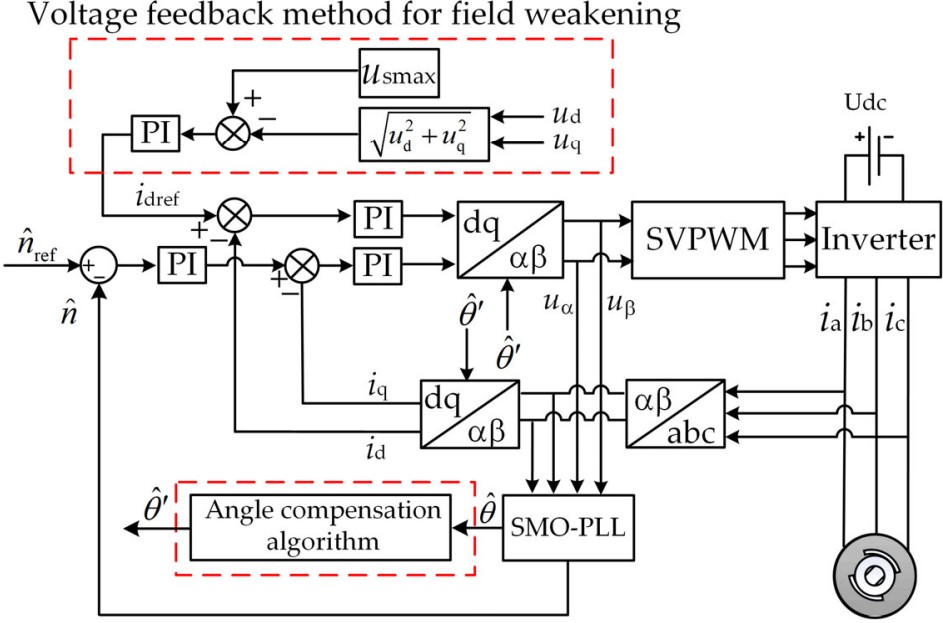

**Figure 3.** Block diagram of sensorless control for high-speed permanent magnet synchronous motor based on discrete current error.

The voltage equation of the surface-mounted permanent magnet synchronous motor in the dq two-phase synchronous rotating coordinate system is:

$$\begin{cases} u_{\mathrm{d}} = Ri_{\mathrm{d}} + L\frac{\mathrm{d}i_{\mathrm{d}}}{\mathrm{d}t} - \omega_{\mathrm{e}}Li_{\mathrm{q}} \\ u_{\mathrm{q}} = Ri_{\mathrm{q}} + L\frac{\mathrm{d}i_{\mathrm{q}}}{\mathrm{d}t} + \omega_{\mathrm{e}}Li_{\mathrm{d}} + \omega_{\mathrm{e}}\psi_{\mathrm{f}} \end{cases} \tag{7}$$

where $u_{\mathrm{d}}$, $u_{\mathrm{q}}$, $i_{\mathrm{d}}$, and $i_{\mathrm{q}}$ are the stator voltage and current in the rotating coordinate system, and $\psi_{\mathrm{f}}$ is the motor rotor flux.

From Equation (3), it can be seen that the SMO method used in this article does not involve the rotor flux linkage term, so only the *d*-axis current needs to be analyzed. Obtain the discretized *d*-axis current by the Euler approximation method:

$$\begin{aligned} i_{\mathrm{d}}(k) &= (1 - \tfrac{T_{\mathrm{s}}R}{L})i_{\mathrm{d}}(k-1) + T_{\mathrm{s}}\omega_{\mathrm{e}}(k-1)i_{\mathrm{q}}(k-1) \\ &\quad + \tfrac{T_{\mathrm{s}}}{L}u_{\mathrm{d}}(k-1) \end{aligned} \tag{8}$$

At this time, assuming that the motor parameters are accurate and the rotor position angle has no error, there should be no obvious static difference between the discrete *d*-axis current and the reference value of the *d*-axis current. When using the sensorless control method, if the estimated rotor position angle contains the error component in Section 2.2, the current equation of the $\gamma$-axis discretization in the estimated rotating coordinate system can be expressed as:

$$\begin{aligned} i_{\gamma}(k) &= (1 - \tfrac{T_{\mathrm{s}}R}{L})i_{\gamma}(k-1) + T_{\mathrm{s}}\hat{\omega}_{\mathrm{e}}(k-1)i_{\delta}(k-1) \\ &\quad + \tfrac{T_{\mathrm{s}}}{L}u_{\gamma}(k-1) + \omega_{e}(k-1)\varphi_{\mathrm{f}}\sin\Delta\theta \end{aligned} \tag{9}$$

In the Formula, $\omega\psi_{\mathrm{f}}\sin\Delta\theta$ is the component of the back electromotive force on the $\gamma$-axis, and $\Delta\theta$ is the angle error analyzed above. It can be seen from the above formula that at this time, the $\gamma$-axis current obtained by Euler's approximation method has a component $e(k-1)\sin\Delta\theta$ formed by the angle error. It is precisely because of the existence of this component that there is a static current difference between $i_{\gamma}$ and the current reference value $i_{\mathrm{dref}}$, and the expression of the current static difference at time $k$ is:

$$\Delta i_{\gamma}(k) = \frac{T_{\mathrm{s}}}{L}e(k-1)\sin\Delta\theta \tag{10}$$

Since $\Delta\theta$ is small, $\sin\Delta\theta \approx \Delta\theta$, and Formula (10) can be changed into Formula (11):

$$\Delta i_{\gamma}(k) = \frac{T_{\mathrm{s}}}{L}e(k-1)\Delta\theta \tag{11}$$

It can be seen from the above Formula that $\Delta i_{\gamma}(k)$ should be equal to 0 when there is no error in the position angle and the motor parameters are accurate. Therefore, it is feasible to use this static current difference value as the basis for evaluating the rotor angle error.

### 3.2. Principle of Angle Compensation Method Based on Discrete Current Error

It can be seen from Equation (11) that when the motor speed is constant, the coefficient of $\Delta\theta$ is a constant. It can be seen that there is an approximately linear relationship between the rotor angle estimation error and the current static error. At this time, the PI controller is used to calculate $\Delta\theta$, with $i_{\mathrm{dref}}$ as the given value of the PI controller and $i_{\gamma}(k)$ as the feedback value of the PI controller. Figure 4 shows the principle diagram of position error compensation.

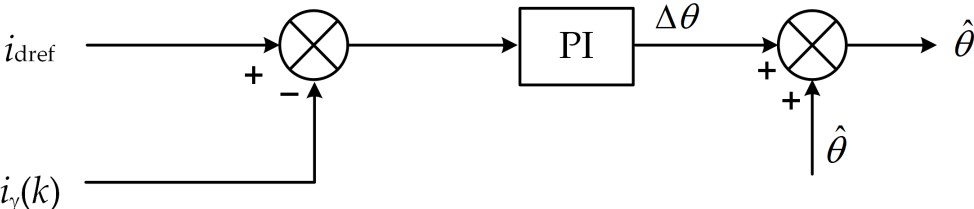

**Figure 4.** Schematic diagram of estimated position angle error compensation.

The compensation angle $\Delta\theta$ is expressed as:

$$\Delta\theta = (k_p + \frac{k_i}{s})(i_{dref} - i_\gamma) \tag{12}$$

where $k_p$ is the proportional coefficient, and $k_i$ is the integral coefficient. If the error value is not zero and the value is large, it indicates that there is an angle error at this time, and the angle compensation algorithm needs to be executed, and then the cycle is repeated until the d-axis discrete error is zero. Compensating the calculated $\Delta\theta$ behind the estimated position angle can compensate for all the errors analyzed in Section 2.2.

*3.3. Variable Proportional Integral Coefficient PI Regulator*

It can be seen from Equation (11) that the coefficient of $\Delta\theta$ is mainly related to the back-EMF, and the value of the back-EMF is related to the motor speed. The higher the speed, the greater the back-EMF, and the larger the back-EMF, the greater the $\Delta i_\gamma$, $\Delta i_\gamma$ is positively correlated with the back-EMF. It can be seen that, for actual working conditions, the relationship between the position angle error and the current static difference is time-varying. Therefore, it is necessary to use a variable proportional integral coefficient PI regulator to ensure that the angle compensation value can be obtained quickly and accurately at different speeds.

The control equation of the traditional linear PI regulator can be expressed in the time domain as:

$$u(t) = k_p e(t) + k_i \int_0^t e(t)\mathrm{d}t \tag{13}$$

where $u(t)$ is the output of the PI regulator, $k_p$ and $k_i$ are the proportional and integral coefficients of the regulator, respectively, and $e(t)$ is the deviation signal. Corresponding to this article, $u(t)$ is the position angle error value $\Delta\theta$, and $e(t)$ is the current static difference value $\Delta i_\gamma$.

The $k_p$ and $k_i$ parameters have different correction effects on the deviation. Increasing the value of $k_p$ can improve the response speed and reduce the steady-state deviation, but if $k_p$ is too large, the overshoot will increase; the $k_i$ parameter is mainly used to eliminate the static error, $k_i$ the larger the value, the smaller the static difference, but if $k_i$ is too large, the transition process of the system becomes longer. From this, the following adjustment mechanism can be determined, that is, $k_p$ is positively correlated with the current static difference $\Delta i_\gamma$. When the error $\Delta i_\gamma$ is large, the larger $k_p$ value should be selected, and when the error is small, the smaller $k_p$ value should be selected, so choose The corresponding positive correlation function corrects $k_p$; while $k_i$ is negatively correlated with the static current difference value. When the error is large, choose a smaller value of $k_i$, and when the error is small, choose a larger value of $k_i$, so you need to choose the corresponding negative value. The correlation function modifies $k_i$. The positive correlation function and negative correlation function used in this article are shown in Equation (14). Figure 5 shows the graphs of these two functions.

$$y_p = 1 - \exp(-0.8x^2)$$
$$\tag{14}$$
$$y_i = \exp(-0.8x^2)$$

The $k_p$ and $k_i$ values are corrected by the selected function, and the rebuilt $k_p$ and $k_i$ function expressions are:

$$k_p = k_{po}\{1 + k_1[1 - \exp(-0.8e^2)]\}$$
$$k_i = k_{io}k_2 \exp(-0.8e^2)$$

(15)

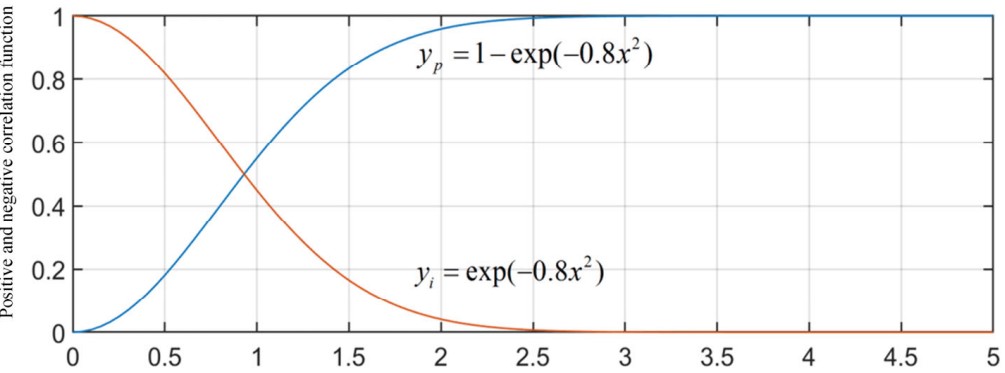

**Figure 5.** Adjusting the positive and negative correlation functions of $k_p$ and $k_i$.

In the Formula, $k_{po}$ and $k_{io}$ are the initial PI parameters, and $k_1$ and $k_2$ are correction coefficients. Only the $k_1$ and $k_2$ parameters need to be adjusted to achieve fast and accurate variable proportional integral control.

Figure 6 is the histogram of static difference, stabilization time, and overshoot under different $k_1$, and Figure 7 is the histogram of overshoot, stabilization time, and static difference under different $k_2$ when $k_1$ is determined.

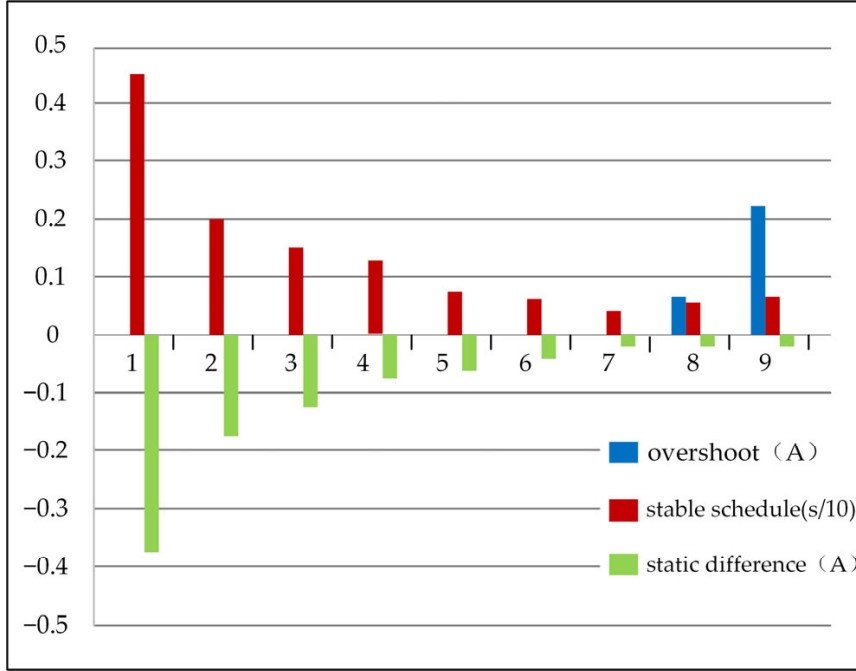

**Figure 6.** Statistical chart of performance data under different $k_1$.

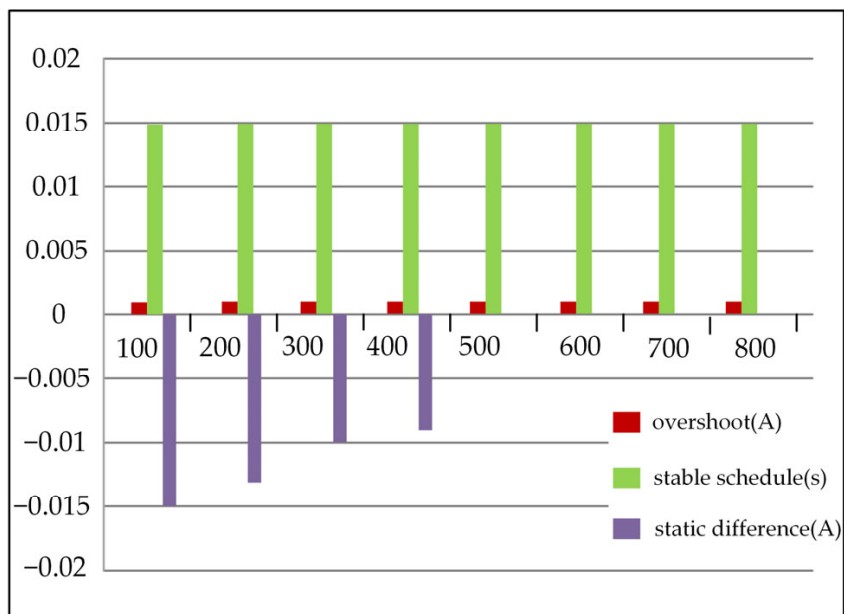

**Figure 7.** Statistical chart of performance data under different $k_2$ when $k_1$ is 7.

It can be seen from Figure 6 that, firstly, the performance under different $k_1$ values is analyzed. From the perspective of static difference, with the increase of $k_1$, the static difference is gradually smaller, but the reduction range is also smaller. When $k_1$ is 7, 8, and 9, the static difference is almost the same, but the overshoot will increase after exceeding 7, so the $k_1$ value is determined to be 7. However, there is still a static error at this time. After determining $k_1$, the appropriate $k_2$ value is selected from Figure 7 to continue to eliminate the static error. The static error gradually decreases with the increase of $k_2$, and there is almost no static error at 500. The tracking value can be stabilized near the given value, so the $k_2$ value is determined to be 500.

### 3.4. Parameter Robustness Analysis of the Proposed Method

Since the high-speed sensorless control algorithm is designed based on the fundamental wave model of the motor, the performance of the algorithm depends on the motor parameters to a certain extent. There is often a certain error between the nominal value and the actual value of the motor, so it is necessary to analyze the parameter robustness of this angle compensation method.

When the motor inductance parameters are mismatched, Formula (8) is rewritten as:

$$i_\gamma(k+1) = (1 - \frac{T_s R}{mL})i_\gamma(k) + T_s \hat{\omega}_e i_\delta(k) + \frac{T_s}{mL}u_\gamma(k) \tag{16}$$

In the Formula, m is the multiple of inductance change. The maximum inductance change rate in this article is 20%, that is, the value of m is between 0.8–1.2.

After the parameter mismatch, Equation (11) will also change. Because the discrete current equation obtained by the Euler approximation method is used, in addition to the position angle error, the other two caused by the inductance parameter mismatch will be added, and the specific expression is:

$$\Delta i_\gamma = \frac{m-1}{mL}T_s R i_\gamma(k) - \frac{m-1}{mL}T_s u_\gamma + \frac{T_s}{mL}e(k)\Delta\theta \tag{17}$$

After sorting out, this Formula can be simplified to:

$$\Delta i_\gamma = \frac{m-1}{mL}T_s[R i_\gamma(k) - u_\gamma] + \frac{T_s}{mL}e(k)\Delta\theta \tag{18}$$

The simultaneous formula can get the following Formula:

$$\Delta i_\gamma = -\frac{m-1}{m} T_s \omega_e i_\delta + \frac{T_s}{mL} \omega_e \varphi_f \Delta\theta \tag{19}$$

Multiply both sides of the equation $m/(T_s \omega_e)$:

$$\frac{m}{T_s \omega_e} \Delta i_\gamma = (1-m)i_\delta + \frac{1}{L} \varphi_f \Delta\theta \tag{20}$$

Since the experimental conditions discussed in this article are for high-speed conditions, the value of $i_\delta$ is relatively small. When there is a 20% inductance parameter mismatch, that is, the value of 1-m is ±0.2, and the absolute value of the first term in Equation (20) is less than 2. Taking the motor used in this article as an example, the inductance value is 3 mH, the rotor flux linkage value is 0.15 Wb, the number of pole pairs of the motor is 2, and $\Delta\theta$ is 1.5 $T_s\omega_e$ when only the digital control delay is calculated. When the speed is equal to 9000 r/min, the angle error value is 0.353 rad at this time. At this time, the value of the second term of Equation (20) is at least 17.65, which is much larger than the first term, so the first term of Equation (20) can be ignored. The proposed method is less affected by the misalignment of motor parameters and works well at 0.8 times inductance mismatch and 1.2 times inductance mismatch with strong parameter robustness.

## 4. Experimental Verification and Result Analysis

### 4.1. Experimental Platform Parameters

The specific parameters of the motor selected in the experiment in this article are shown in Table 1.

**Table 1.** Motor rated parameter table.

| $P_N$/kW | $T_N$/N·m | $\omega_N$/r/min | $J$/kg·m² | $L_d$, $L_q$ /mH | $n_p$ | $R_s$/Ω | $\psi_f$/Wb |
|---|---|---|---|---|---|---|---|
| 3.7 | 11.8 | 3000 | 0.0012 | 3 | 2 | 0.38 | 0.15 |

Because the method proposed in this paper is mainly aimed at the working condition of the motor in the middle and high-speed stage, at this time, limited by the switching performance of the power device, the motor system will be in the case of the low ratio of switching frequency to motor rotor frequency, that is, high-speed and low carrier ratio state. If the sampling and updating method described in Section 2.2.2 is still used at this time, the system will produce a greater digital control delay. Therefore, this paper adopts the control mode of double sampling and double update, that is, in addition to sampling and algorithm execution at the beginning of each switching cycle, sampling and algorithm execution are also performed at $1/2$ $T_{PWM}$ time of each switching cycle. Therefore, after adopting the double sampling and double update mode, the sampling delay is shortened to 0.5 $T_{PWM}$, the PWM update delay is shortened to 0.25 $T_{PWM}$, and the total control delay of the system is 0.75 $T_{PWM}$. This control mode can effectively reduce the impact of digital control delay, especially in high-speed and low carrier wave ratio conditions can effectively improve the quality of the three-phase current waveform.

In this paper, the control mode of double sampling and double updating is selected. The IGBT switching frequency is limited to 4 kHz, and the control frequency is selected as 8 kHz. The motor speed range is 0–9000 r/min, and the voltage feedback field weakening control is performed near the 7000 r/min speed. Figure 8 is the experimental system platform diagram.

Inverter          Main control board

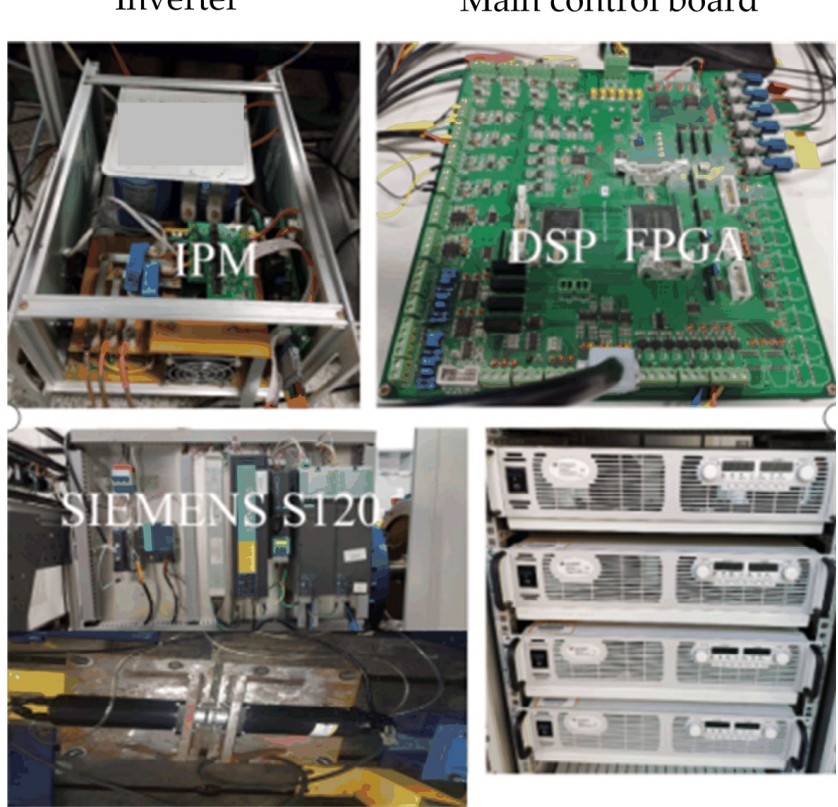

Experimental motor Load
motor          DC power supply

**Figure 8.** Experimental system.

*4.2. Experimental Analysis*

Figure 9 shows the speed control obtained by the traditional back EMF method test waveform. Figure 9a shows the speed waveform, $i_d$, and $i_q$ waveforms of the motor from 1500 r/min to 6700 r/min.

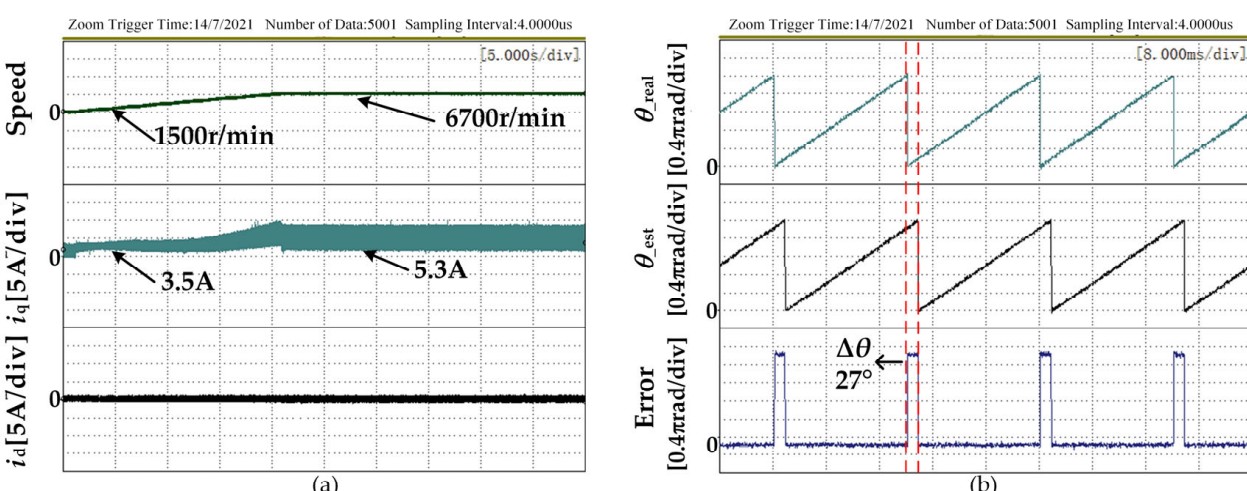

**Figure 9.** *Cont.*

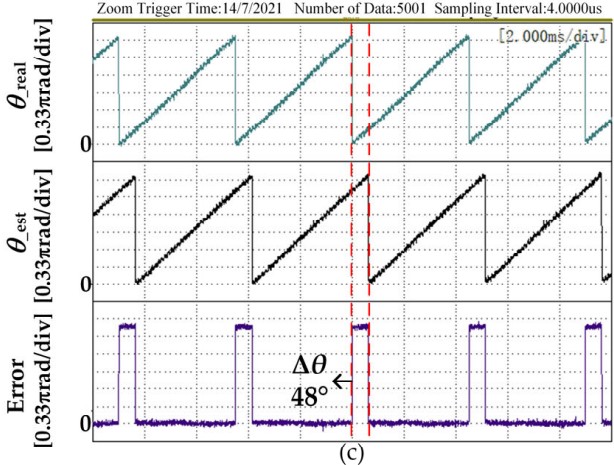

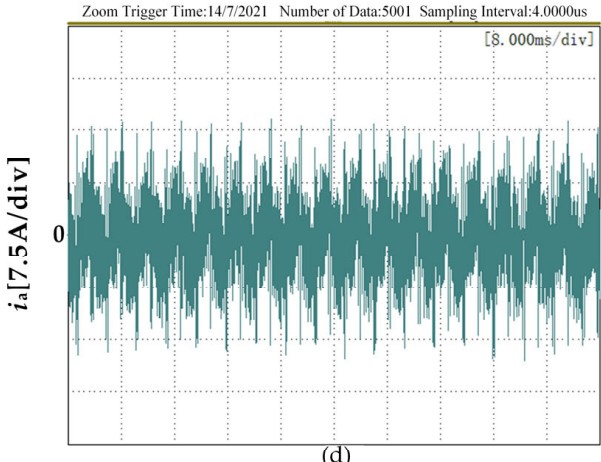

**Figure 9.** The experimental results with the traditional method: (**a**) Speed and *dq* axis current when the speed is increased from 1500 r/min to 6700 r/min; (**b**) Estimated position angle, actual position angle and error under 1500 r/min; (**c**) Estimated position angle, actual position angle and error at 6700 r/min; (**d**) Phase A current at 6700 r/min.

Figure 9b,c are the estimated rotor position angle, resolver output angle, and their errors at 1500 r/min and 6700 r/min respectively; Figure 9d is the stator current of phase A at 6700 r/min. It can be seen from Figure 9b,c that as the speed increases, the rotor position angle error obtained by the traditional back EMF method increases significantly, and the hysteresis phenomenon becomes more obvious. Because of this, during the experiment, the maximum speed can only be controlled to 6700 rpm when using the traditional method, and the motor will be completely out of control after the speed is exceeded.

Figure 10 shows the experimental results when the motor parameters are accurate using the method proposed in this article. Figure 10b,c are the estimated rotor position angle, resolver output angle, and the errors of the motor at 6700 r/min and 9000 r/min, respectively. It can be seen from the figure that when the motor is at 6700 r/min, the rotor position and angle error estimated by the proposed method is significantly smaller than the position angle error under the traditional method shown in Figure 9c. At the same time, when the motor is at 9000 r/min, the rotor angle error estimated by the method proposed in this article does not increase.

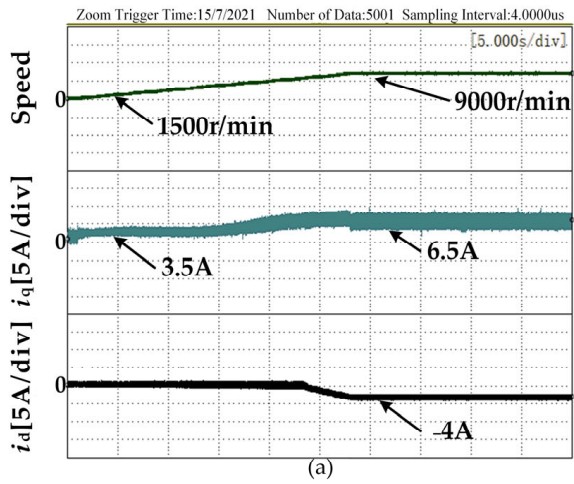

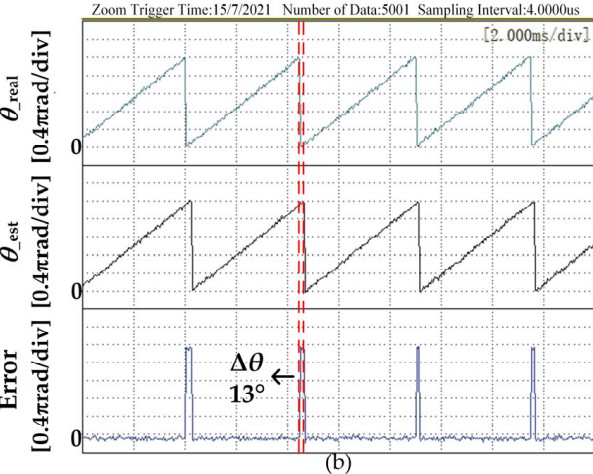

**Figure 10.** *Cont.*

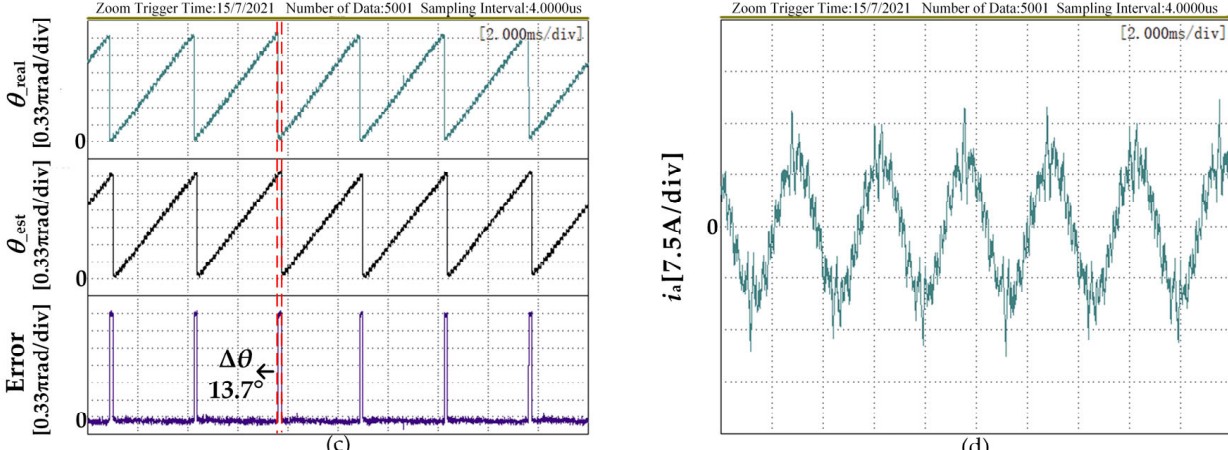

**Figure 10.** The experimental results with the proposed method: (**a**) Speed and *dq* axis current when the speed is increased from 1500 r/min to 9000 r/min; (**b**) Estimated position angle, actual position angle and error under 6700 r/min; (**c**) Estimated position angle, actual position angle and error at 9000 r/min; (**d**) Phase A current at 9000 r/min.

In order to prove that the method proposed in this paper has better parameter robustness, Figures 11 and 12 are the experimental results of the motor inductance parameters equal to 0.8 times and 1.2 times the actual inductance parameters used in the program, respectively.

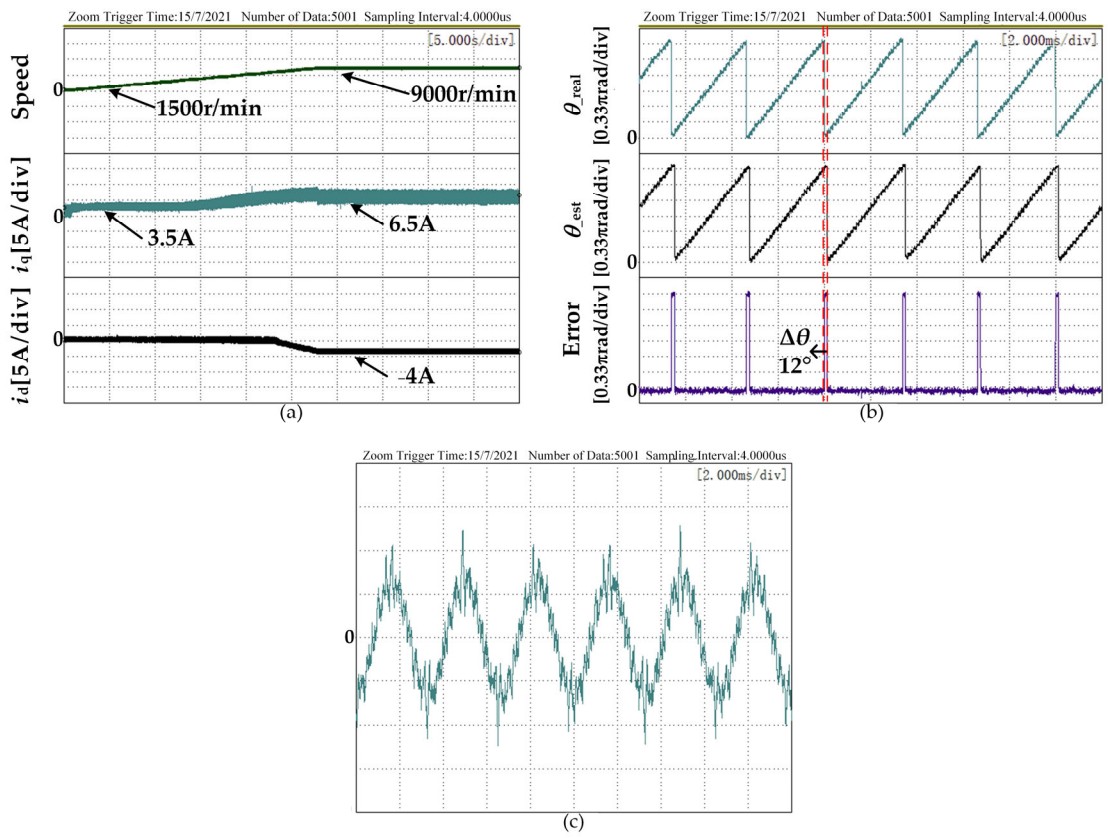

**Figure 11.** The speed-up experimental results with the proposed method when the program parameter is equal to 0.8 times the nominal inductance: (**a**) Rotating speed and *dq* axis current at 1500 r/min to 9000 r/min; (**b**) Estimated position angle, actual position angle and error at 9000 r/min; (**c**) A-phase current at 9000 r/min.

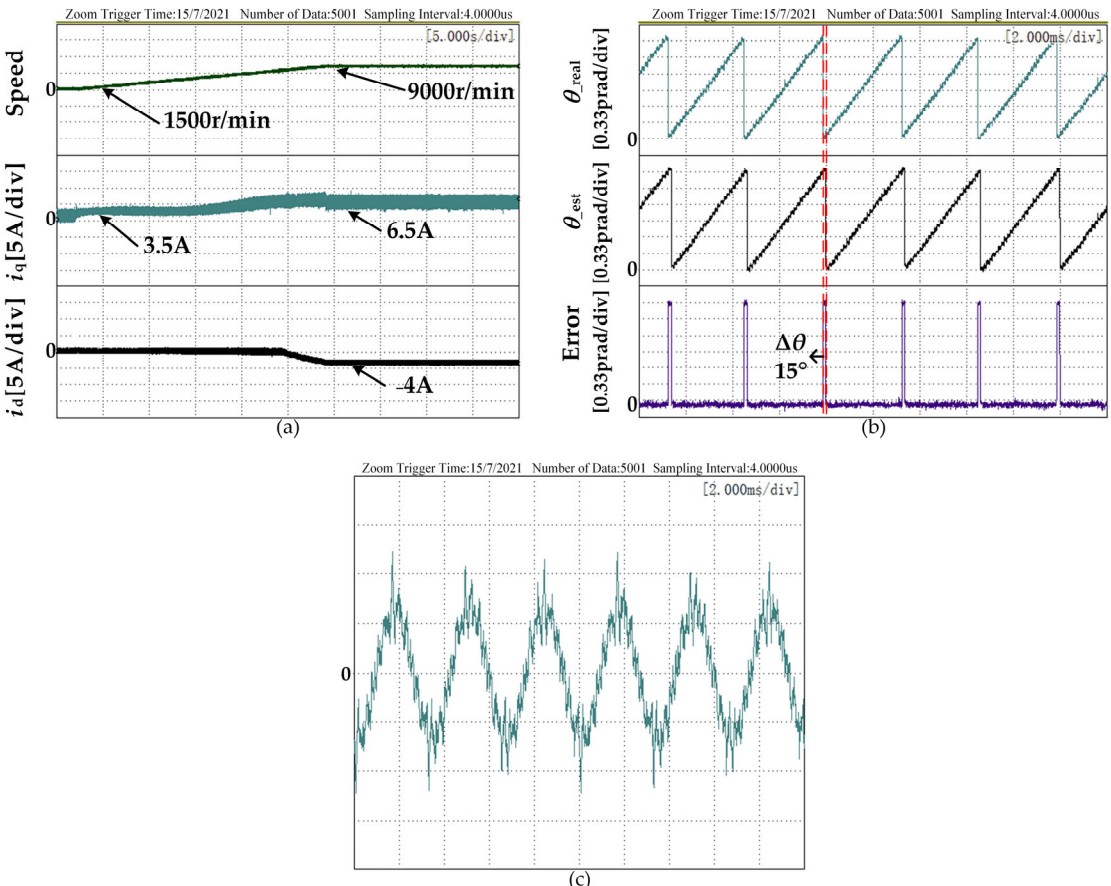

**Figure 12.** The speed-up experimental results with the proposed method when the program parameter is equal to 1.2 times the nominal inductance: (**a**) Rotating speed and *dq* axis current at 1500 r/min to 9000 r/min; (**b**) Estimated position angle, actual position angle and error at 9000 r/min; (**c**) A-phase current at 9000 r/min.

It can be seen from Figures 11b,c and 12b,c that although the inductance parameters and actual parameters are in the actual control program. There is a deviation, but the estimated rotor position angle and its error with the actual position angle do not change significantly.

## 5. Conclusions

This article proposes a sensorless control method for permanent magnet synchronous motors for high-speed and carrier wave ratio conditions. Compared with other high-speed low-carrier-ratio position error compensation methods, the proposed method has the advantages of a simple compensation mechanism, superior compensation effect, and high compensation accuracy, and has stronger stability and reliability when applied to the speed-up process of electric vehicles. The specific summary is as follows:

(1) The proposed method uses the static difference of the d-axis current discretized by Euler's approximation method as the basis for judging whether there is an angle estimation error, that is when there is an angle estimation error, there is a static difference between the actual value of the d-axis current and the reference value, and otherwise, there is no error.

(2) Since the static difference value of the d-axis current changes with the speed change, a PI controller with variable proportional and integral coefficient is required, that is, the input value of the PI controller is the given value of the d-axis current, and the feedback value is the discretized d-axis current, The output value is the position angle error compensation amount. By using different proportional integral coefficients at different speeds, it is ensured that the position angle can be accurately and quickly

estimated using the proposed sensorless control method at different speeds, and the estimation error is unchanged.

(3)  Through theoretical analysis and experimental verification, it can be seen that the proposed method can accurately compensate the estimated rotor position in the case of high-speed and low carrier wave ratio, improve the control accuracy, and have strong parameter robustness. In the case of the model, mismatch caused inductance parameter changes, the angle error can still be accurately compensated to ensure the motor performance.

**Author Contributions:** Conceptualization, Z.W. and D.D.; methodology, Z.W. and X.L.; software, D.D. and C.L.; validation, D.D.; formal analysis, Z.W. and X.L.; investigation, D.D.; resources, Z.W.; data curation, H.Z. and Q.Y.; writing—original draft preparation, D.D.; writing—review and editing, X.G. and X.L.; visualization, D.D. and L.G.; supervision, Q.Y. and H.Z.; project administration, Z.W. and X.G.; funding acquisition, Z.W. and X.L. All authors have read and agreed to the published version of the manuscript.

**Funding:** This research was funded by "The National Natural Science Foundation of China, grant number 51977150", "The National Natural Science Foundation of China, grant number 52077155", and "Zhejiang Provincial Natural Science Foundation of China, grant number LY22E070005".

**Institutional Review Board Statement:** Not applicable.

**Informed Consent Statement:** Not applicable.

**Data Availability Statement:** Not applicable.

**Conflicts of Interest:** Qi Yu and Haifeng Zhang are employees of Baodi Power Supply Branch of State Grid Tianjin Electric Power Company. The paper reflects the views of the scientists, and not the company.

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
