# Peer review of "Sensorless Control Method of High-Speed Permanent Magnet Synchronous Motor Based on Discrete Current Error"

_wevj, doi:10.3390/wevj14030069_

Round 1

Reviewer 1 Report

The paper expose an interesting research subject in relation to Permanent Magnet Synchronous Motor and the control strategy. The static error between the discrete d-axis current and the matching reference value is used to drive a sensorless control approach in this paper. However, the contributions of this work are not clearly stated. Below are my comments for the improvement of the work: 

1- Please clarify the contributions of this work in the abstract and in the introduction section. The authors claimed at the end of the introduction that ‘'the main factors that lead to the error in the estimated …’' What has been specifically extended ? Please provide more information and explanations.

2- Please improve the language by correcting grammatical problems and typos throughout the work.

3- Please make sure symbols are consistently used throughout the work.

4- How did the author to calculate and compensation angle ? Can this be done using an estimate KALMAN filter ?

5- In Figure 10 the current value is irrational for example in 1500r/min the value of current is not clear.

6-The author should be add a comparison between the method used  and the other methods. And add the effect of this method in controlling the EV.

7- Please improve all figures (quality and clarity of the numbers).

8- The conclusion of this paper should point out the result in a more precise way and show the whole contribution for this research.

Author Response

Thank you for your suggestion, we have modified the mansctript.

Reviewer 2 Report

This paper presents sensorless control method of high speed PMSM based on discrete current error. The idea presented in the paper is good. The article is well organized; the results are well presented and discussed. However, the article needs minor revision. Some comments and suggestions are given here:

1)      The choice of the proposed control should be justified.

2)      The proposed model should be compared with previous works to verify its validity in the paper.

3)      More references to PMSM papers should be included, like:

Implementation and validation of Backstepping control for PMSG wind turbine using dSPACE controller board. Energy Reports, 2019, vol. 5, p. 807-821.

Model reference adaptive system based DPC-SVM control for permanent magnet synchronous generator. In : Digital Technologies and Applications: Proceedings of ICDTA’22, Fez, Morocco, Volume 1. Cham : Springer International Publishing, 2022. p. 535-544.

Nonlinear backstepping control for PMSG wind turbine used on the real wind profile of the DakhlaMorocco city. International Transactions on Electrical Energy Systems, 2020, vol. 30, no 4, p. e12297.

4)      Prepare the nomenclature table for your abbreviations

5)      Interpretation of results and study conclusions need to be improved.

Author Response

(The authors gave the same response as above.)

Round 2

Reviewer 1 Report

The paper is better than the previous version and the author has answered most of the questions. However, state of the art must be updated by some references in the field. 

- Revise figure 3 and improve the quality.

- Spacing should be checked throughout the paper.